# Optimized Extraction of Bioactive Polysaccharides from Wild Mushrooms: Toward Enhanced Yield and Antioxidant Activity

**DOI:** 10.3390/molecules30234647

**Published:** 2025-12-03

**Authors:** Aya Samy Ewesys Khalil, Marcin Lukasiewicz

**Affiliations:** 1Department of Food Engineering and Machinery for Food Industry, Faculty of Food Science, University of Agriculture in Krakow, al. Mickiewicza 21, 31-120 Krakow, Poland; aya.samy@agr.cu.edu.eg; 2Food Science Department, Faculty of Agriculture, Cairo University, Giza 12613, Egypt

**Keywords:** wild mushroom, bioactive polysaccharides, optimization, aqueous extraction, *Suillus luteus*, *Tricholoma equestre*, *Hydnum repandum*, antioxidant activity

## Abstract

The aqueous extraction of bioactive compounds from wild mushroom fruiting bodies, focusing on carbohydrates, has been systematically examined. This study includes three mushroom species common in the northern hemisphere: *Suillus luteus*, *Tricholoma equestre*, and *Hydnum repandum*. These species were selected for their potential as cost-effective sources of bioactive compounds. For each species, the optimization of liquid-to-solid (LS) ratio (50:1 *v*/*w* to 150:1 *v*/*w*), temperature (70–90 °C), and processing time (3 to 5 h) was conducted to determine optimal parameters for total carbohydrate content (TCC), while minimizing reducing sugars to favor higher molecular weight polysaccharides. The bioactive properties were explored and optimized based on the antioxidant properties of extracts. The data were compared with previous studies on commonly cultivated mushrooms, such as *Agaricus bisporus*. Results show that the high LS ratio has the most significant influence on TCC content, though optimal values for other parameters (temperature and time) vary by species. The optimal LS ratios were 150:1 for *Suillus luteus*, 149.89 for *Tricholoma equestre*, and 149.76 for *Hydnum repandum*. Temperature and duration varied among species, with *Suillus luteus* requiring 5 h at 89.92 °C, *Tricholoma equestre* needing 3.98 h at 70.07 °C, and *Hydnum repandum* requiring 3.00 h at 70.01 °C. A similar trend was observed in minimizing reducing sugars, confirming the high LS ratio may support extracting longer polysaccharide chains. Studies on antioxidant activity revealed that bioactive molecules in the extract are water-soluble molecules; however, the optimal values for antioxidant activity are strongly mushroom-species-dependent. The optimal conditions for enhancing antioxidant activity of aqueous extracts, measured by ABTS method, were: for *Suillus luteus*, an LS ratio of 123.68, 5 h, and 86.63 °C; for *Tricholoma equestre*, an LS ratio of 95.27, 4.05 h, and 73.87 °C; and for *Hydnum repandum*, an LS ratio of 50.01, 5 h, and 89.98 °C. The aqueous extraction method proved efficient for recovering bioactive polysaccharide fractions from wild mushrooms.

## 1. Introduction

In contemporary times, there is a pressing necessity to identify new sources of nutritional and bioactive substances to support the growing human population and ensure their health is maintained [1,2,3]. Climate change further complicates this endeavor, posing challenges to the utilization of traditional plant and animal sources for food supplies [4]. Fungi, representing the third domain of life on Earth, have been utilized for both medicinal and nutritional purposes for centuries and play a crucial role in maintaining balanced ecosystems globally. They are abundant in bioactive compounds and essential nutrients, including proteins, carbohydrates, dietary fiber, minerals, and vitamins, while maintaining a low-fat content. The diversity of edible mushrooms encompasses hundreds of species; however, only a limited number are currently under cultivation. The global mushroom market was valued at USD 63 billion in 2024, with cultivated edible mushrooms comprising 54% of this market, medicinal mushrooms 38%, and wild mushrooms 8% [5]. Advances in mycoscience suggest that numerous mycorrhizal wild mushroom species may soon transition from the “wild mushroom” category to the “cultivated” category [6,7]. In numerous countries, hundreds of tons of wild mushrooms are harvested annually, entering the HoReCa system, commercial trade, or being processed by individual consumers. This underscores the significance of wild mushrooms as a vital source of bioactive nutrients and a promising foundation for innovative food products within the framework of a sustainable and circular economy.

Carbohydrates, encompassing mono-, oligo-, and polysaccharides, are integral to human nutrition, serving as energy sources and functional components for the microbiome, in addition to fulfilling numerous specific roles within the human body [8,9]. Mushrooms, noted for their distinctive nutritional profiles, have been demonstrated to contain 50 to 65% total carbohydrates, which are essential for various physiological functions [10]. These carbohydrates predominantly consist of bioactive polysaccharides, such as beta-glucans, which are acknowledged for their health-enhancing properties, including immune modulation and glycemic regulation [11]. Research on wild mushrooms has predominantly concentrated on their chemical composition, quality, safety, and medicinal applications [12,13]. However, data regarding their potential as a technological source of valuable nutrients remain limited. In contrast, studies on cultivated species are more comprehensive, addressing various aspects of their composition and their application in innovative food products. This knowledge gap concerning wild species prompted us to investigate the extraction process of the saccharide-rich fraction from three wild mushroom species commonly found in the forests of Central and Eastern Europe. Building upon our prior research on the commonly cultivated mushroom *Agaricus bisporus* [14], the current study extends the investigation to wild species, specifically *Suillus luteus* (commonly referred to as slippery jack or sticky bun), *Tricholoma equestre* (L.) P. Kumm. (known as man on horseback or yellow knight), and *Hydnum repandum* L. (commonly called sweet tooth, pig’s trotter, wood hedgehog, or hedgehog mushroom). Foraging for wild mushrooms is a prevalent outdoor activity among individuals in Europe, particularly within Slavic regions. Various species are recognized for their distinct culinary values, such as Boletus edulis, Morchela esculenta, and different types of truffles. Conversely, other species, including those examined in this study, are considered less valuable, resulting in significantly lower market prices. Consequently, these species may serve as a promising alternative source of polysaccharides for non-culinary applications in food processing. The selection of mushroom species for extraction was also informed by their diverse fruiting body structures and chemical compositions [13].

*Suillus luteus* is an ectomycorrhizal fungus that establishes symbiotic associations with various pine tree species, facilitating its extensive distribution across the Northern Hemisphere and regions of the Southern Hemisphere, such as Patagonia [15]. In European forests, this mushroom is frequently harvested, although it is deemed less valuable compared to other species like *Boletus edulis*, etc. *Suillus luteus* is edible and is often traded as a food commodity in numerous countries. *Suillus luteus* is esteemed for its substantial antioxidant capacity and antimicrobial properties, rendering it significant both ecologically and for health [16]. It is also abundant in essential vitamins and minerals; however, these attributes can be considerably diminished during culinary processing, thereby affecting its nutritional value [17].

*Tricholoma equestre* is a wild mushroom distinguished by its unique morphological and ecological attributes, rendering it a notable member of the *Tricholoma* genus. This species is predominantly located in coniferous and mixed forests throughout Europe and North America, where it forms ectomycorrhizal associations with trees, thereby facilitating nutrient exchange and supporting forest ecosystems [18]. Morphologically, *Tricholoma equestre* is characterized by its compact, fleshy structure, featuring a thick stipe and a distinctive yellow hue, which contributes to its common name [19]. Despite its popularity for its palatable taste and culinary applications, the mushroom has become contentious due to its potential toxicity, which is sometimes associated with rhabdomyolysis [20]. However, there is no definitive conclusion regarding the mushroom’s toxicity. In Europe, *Tricholoma equestre* is classified as an edible mushroom in 18 countries, while in 19 countries, it is deemed non-edible. Despite these risks, *Tricholoma equestre* remains a subject of scientific investigation due to its ecological significance and complex biochemical profile. As research progresses, understanding the conditions under which *Tricholoma equestre* can be safely consumed is crucial for mitigating health risks and harnessing its nutritional benefits [21]. Given the controversies, the mushroom is regarded as having low value and interest; thus, it may serve as a suitable raw material for the extraction of safe polysaccharide fractions.

*Hydnum repandum* is a mushroom notable for its substantial ecological contributions and nutritional benefits for humans [22]. Nevertheless, even in regions where the collection of wild mushrooms is prevalent, such as Central and Eastern Europe, this species is infrequently harvested. This is primarily due to the presence of other species that are considered to be more palatable and widely favored [23]. As an ectomycorrhizal fungus, it establishes symbiotic relationships with trees, facilitating nutrient exchange and playing a crucial role in enhancing forest ecosystem biodiversity [22]. From a nutritional perspective, *Hydnum repandum* serves as a low-calorie source of dietary fiber, protein, and essential minerals, complemented by its notable antioxidant properties. Research has emphasized its high phenolic content, which enhances its efficacy as a dietary antioxidant [24]. The presence of bioactive compounds, such as protocatechuic acid, further highlights its health benefits, as these metabolites offer protective effects against oxidative stress [25]. *Hydnum repandum* is also recognized for its medicinal potential [26]. The synergistic effects of its bioactive compounds render it a promising candidate for pharmaceutical applications, as it can interact beneficially with chemotherapeutic agents [27].

Building upon our prior research on cultivated mushrooms (*Agaricus bisporus*) [14], this study aims to optimize the extraction of polysaccharide-rich fractions from *Suillus luteus*, *Tricholoma equestre*, and *Hydnum repandum*. The objective is to determine whether these wild mushrooms can serve as valuable sources of polysaccharide fractions with functional antioxidant properties. While cultivated mushroom species are produced for their nutritional or medicinal benefits, many wild mushroom species may offer polysaccharides that can be utilized in the development of novel food products with specific functionalities, either nutritional or technological.

The goal was achieved through the application of a Response Surface Methodology design, which elucidates the relationship between three factors and various response variables to determine the optimal conditions for polysaccharide extraction. Specifically, the factors under investigation include the water-to-mushroom ratio, extraction temperature, and extraction time. The outcomes of this optimization are compared with those obtained from optimization based on antioxidant properties, measured using different methods, to elucidate the potential mechanism underlying the bioactivity of the extracts.

## 2. Results and Discussion

Mushrooms are significant sources of bioactive compounds, with carbohydrates (mono-, oligo-, and polysaccharides) being among the most crucial groups. These carbohydrates constitute approximately 35–75% of the dry matter of mushroom fruiting bodies [28]. The carbohydrates encompass various polysaccharides, such as β-glucans, chitin, glycogen, and mono- and disaccharides, which hold nutritional and health importance [29].

The quantity of extracted polysaccharides in wild mushrooms was substantially lower than that in the previously studied *Agaricus bisporus* [14] and was independent of the mushroom species. Among the mushrooms examined, *Suillus luteus* exhibited the highest polysaccharide content (approximately 28.5 mg/g), whereas *Trichloloma equestre* showed the lowest value (approximately 12 mg/g). These findings indicate that the carbohydrate content in the fruiting body is not the sole determinant of the carbohydrates present in the extract. Additional factors include the water solubility of higher molecular compounds (polysaccharides) and the internal structure of the fruiting body cells, specifically the interaction of carbohydrates with other molecules, including proteins [30]. Such macromolecular superstructures can significantly reduce the solubility of polysaccharides and some oligosaccharides.

The results obtained in this study, when compared with similar findings, indicate that the total carbohydrate content in the optimized environment was more effective than that recorded for other wild mushrooms, such as *Coprinopsis atramentaria*, *Lactarius bertillonii*, *Lactarius vellereus*, and *Xerocomus chrysenteron*. In these species, the extracted carbohydrate content did not exceed 17 mg/g and was often below 10 mg/g [31]. Conversely, it has been reported [32] that for other mushroom species, including *Armillaria mellea*, *Calocybe gambosa*, *Clitocybe odora*, and *Coprinus comatus*, prolonged aqueous extraction of freeze-dried material resulted in a carbohydrate content higher than that presented in our research, yet still lower than that obtained for *Agaricus bisporus* in our previous work [14]. The observed differences between the studies can be attributed to variations in the chemical composition of the fruiting bodies of different mushroom species and the extraction processes employed, such as the use of pretreatment, dried or freeze-dried raw material, and other parameters that were observed for both wild and cultivated species in the group of edible and medicinal mushrooms [33].

The optimization of the polysaccharide extraction process was conducted using second-order polynomial equations, resulting in a model that exhibited significant alignment with the experimental data. The models achieved an R^2^ value exceeding 0.85 for the total carbohydrate content in the sample, as illustrated in Table 1. In the model, a positive beta coefficient signifies a direct relationship between the explanatory variable and the outcome, whereas a negative coefficient denotes an inverse relationship. The ANOVA results for the examined response variables, presented in the Appendix A, indicate that the model parameters effectively account for the experimental variation in the response variables.

The optimization of total carbohydrate content (TCC) transferred from the fruiting bodies into the extract, as described in our preliminary studies on *Agaricus bisporus*, demonstrated a strong correlation with the investigated variables, namely time, temperature, and the water-to-mushroom (LS) ratio. In contrast, the optimal values obtained for cultivated mushrooms (*Agaricus bisporus*) differed from those of wild species, with significant variations observed among wild mushroom species. In this context, the common variable exerting a similar influence on carbohydrate concentration in the extract was the highest LS ratio. The parameter value for all three species was approximately at the upper end of the investigated range (150:1), i.e., 150:1; 149.89 and 149.76, respectively, as shown in Table 2. Conversely, the optimal LS ratio for previously investigated *Agaricus bisporus* was lower, not exceeding 120:1 [14]. This occurrence may be attributed to the different internal structure of polysaccharides in wild mushroom extracts, e.g., a higher hydrodynamical radius or a bigger ratio of branching, necessitating more solvent for effective transfer from tissue to solution. The phenomenon is evident in the regression coefficients, where the β_1_ coefficient, representing the LS ratio, had the highest value among all linear, quadratic, and mixed coefficients, excluding β_13_ (mixed LS ratio and extraction time) coefficient in the case of *Hydnum repandum* and *Suillus luteus*.

Notably, in the case of *Hydnum repandum*, only the linear coefficient influenced TCC in extracts, while for all other coefficients, the *p*-value exceeded 0.01, indicating no statistical significance. Together with the highest R^2^ value for this optimization, it can be concluded that the carbohydrate extraction process may be the simplest among all investigated species in terms of mass transfer. For other species, such as *Suillus luteus* and *Tricholoma equestre*, the mass transfer is more complex and resembles that observed for *Agaricus bisporus*; however, differing regression coefficients result in distinct optimal values for each species. For *Suillus luteus*, the optimal conditions for all investigated variables were also at the upper edge of the proposed range, as reflected in the regression coefficient, which consistently exhibited positive values regardless of the coefficient type. This phenomenon was not observed for any other species, including both wild and cultivated mushrooms.

On the other hand, the differences in time and temperature parameters were observed for all three investigated species. Those differences cover the whole investigated range. The observed phenomenon needs detailed investigation; however, its explanation is probably differences in fruiting body tissue structure, including, e.g., the cell wall thickness, mechanical properties, as well as its permeability for water and water solutions.

The total carbohydrate content (TCC) in the obtained extracts reflects the quantity of water-soluble carbohydrates that can be extracted from the fruiting body of a mushroom. Equally significant is the degree of polymerization of these carbohydrates. Bioactive carbohydrate-based compounds in mushrooms predominantly possess macromolecular structures, encompassing various glucans (α and β) as well as chitosan and its derivatives [34]. However, chitin-based materials are characterized by their poor solubility and cannot be extracted using water as a solvent [35]. Consequently, optimization was conducted to reduce the simple sugar content in the obtained extracts. In instances where the extract exhibits a high TCC alongside a low reducing sugar (RS) content, the molecules in the extract possess a polymeric structure with only one reducing aldehyde group present in an extended polysaccharide chain. The optimization of RS in extracts was achieved by minimizing the RS value, thereby maximizing the extraction of polysaccharide macromolecules from the substrate. Compared to the previously studied cultivated mushroom, *Agaricus bisporus* [14], the overall RS values for the three wild mushroom species were significantly higher, reaching 6.2 µmol/mL of free aldehyde groups in the case of *Tricholoma equestre*. Additionally, the value observed for *Hydnum repandum* was substantial, reaching 6 µmol/mL. The lowest RS parameter was observed for *Suillus luteus*; however, this value was more than twice that of *Agaricus bisporus* [14]. This phenomenon is likely attributable to the fact that commercial cultivation, as in the case of *Agaricus bisporus*, occurs under constant and optimized conditions, allowing the fruiting body to develop without environmental interference, such as temperature or humidity fluctuations [36]. In the case of wild mushrooms, the growth of the fruiting body takes place in constantly changing conditions. Consequently, the polysaccharides synthesized during the growth of *Agaricus bisporus* fruiting bodies can achieve a higher molecular mass, resulting in a lower value of reducing sugars.

In contrast, focusing on the three investigated wild mushroom species, it is noteworthy that in the case of *Suillus luteus*, the lowest RS value was accompanied by the highest TCC parameter. This clearly demonstrates the macromolecular nature of the carbohydrates extracted from this mushroom. This may also corroborate the known viscid coating secretion of mushrooms containing high macromolecular polysaccharides with the ability to form a hydrocolloid network in the fruiting body tissues, protecting the fruiting body of *Suillus luteus* against drying out.

For the other two species, the molecular structure of the extracted carbohydrates indicated the presence of shorter polymer chains (higher RS values). The appearance of the mushroom after water contact confirms its reduced ability to bind water, as the surface of the mushroom became significantly less sticky (or “slippery”).

The optimization of the reducing sugar (RS) content in the extracts revealed a complex mechanism of mass transfer involving compounds with varying polymer chain lengths. The optimization of the lowest RS content across all mushroom types was conducted using second-order polynomial equations, resulting in a model that aligns with the experimental data with a high degree of significance. Regression analysis of the three mushroom species demonstrated distinct effects of the three factors—temperature, time, and solid/liquid ratio—on the reducing sugar content (Table 3). The models achieved an R^2^ value exceeding 0.84 for the reducing sugar content in the samples, except for *Hydnum repandum*, which achieved an R^2^ level of 0.76, only showing weaker model fitting, as illustrated in Table 3. The ANOVA results for the examined response variables, presented in the Appendix A, indicate that the model parameters effectively account for the experimental variation in the response variables.

In the case of *Agaricus bisporus* [14], as well as for all investigated species, most factors (linear, quadratic, and mixed) influenced the obtained RS content. The solid/liquid ratio exhibited the strongest negative linear effect, with noticeable interactions with all parameters. This is a logical consequence of polymer solubility phenomena, where long polymer chains require a greater amount of solvent to remain in solution, primarily as random coils [37]. Notably, for *Suillus luteus* and *Hydnum repandum*, the shortest extraction time is preferable to achieve the minimum RS value (Table 4). In contrast, the longest optimized extraction time was observed for *Tricholoma equestre*.

Similar phenomena were observed regarding temperature optimization, which did not exceed 75 °C for *Suillus luteus* and *Hydnum repandum*, while for *Tricholoma equestre*, a much higher temperature was calculated as optimal (approximately 88 °C). This observation indicates that the extraction of polysaccharides from *Tricholoma equestre* involves distinct mechanisms. One possible explanation is the requirement for increased energy to separate the polysaccharide-soluble fraction from the insoluble tissues of the mushroom fruiting body. This process necessitates an extended duration and elevated temperature to dissociate these fractions. Consequently, polysaccharide hydrolysis occurs as a secondary reaction, leading to an increased RS value.

Antioxidants, as bioactive compounds and constituents of food, play a crucial role for consumers by contributing to the maintenance of health and overall well-being [38]. These compounds are present in foods of various origins, including plant, animal, and fungal sources. They may exist as independent food ingredients (low-molecular-weight antioxidant molecules) or be associated with larger macromolecules within food [39]. In the latter scenario, their physical properties, such as solubility, are largely influenced by the characteristics of the macromolecular hosts. In the present study, we concentrated on the extraction of polysaccharides from the fruiting bodies of mushrooms, into which bioactive components, such as antioxidants, can be incorporated. These systems were evaluated using radical scavenging assays, including reactions with the stable radical DPPH, the cation radical ABTS, and OH· radicals generated from hydrogen peroxide solution [40].

The scavenging activities of various mushroom species exhibited significant variation under different conditions. This variability underscores the substantial influence of the LS ratio, temperature, and time on antioxidant activity. A comparison between the studied wild mushroom species and previously examined cultivated species, such as *Agaricus bisporus*, revealed that *A. bisporus* demonstrated the lowest DPPH scavenging activity, approximately 3.5% [14]. In contrast, the wild species exhibited activities three to four times higher, though not exceeding 12.5%. These relatively low DPPH radical scavenging values support our earlier hypothesis that procedures involving the removal of low-molecular-weight compounds with alcohol result in the washing out of not only monosaccharides but also unbound, alcohol-soluble antioxidants, as opposed to water-soluble ones. Despite the low antioxidant activity measured by DPPH in all examined wild mushrooms, it was feasible to optimize the extraction parameters to enhance antioxidant activity. This optimization yielded a model that aligned with the experimental data, with R^2^ values ranging from 0.821 to 0.854 for antioxidant activities across all species, as presented in Table 5. The highest DPPH scavenging activity was observed at approximately 90 °C, 5 h, and a 50 LS ratio for *Suillus luteus*; 90 °C, 3 h, and a 150 LS ratio for *Tricholoma equestre*; and 80 °C, 4 h, and a 50 LS ratio for *Hydnum repandum* (Table 6). The optimal parameters in this context differ slightly from those for TCC; however, it is noteworthy that higher temperatures facilitate an increase in the DPPH scavenging of the extract, likely due to heat-induced hydrolysis of bonds linking the macromolecular host to the antioxidant molecules [41,42]. Additionally, the observed variations in LS ratios for *Tricholoma equestre* may be attributed to the differential solubility of antioxidant moieties extracted from the fruiting bodies of this species.

The range of DPPH scavenging activity reported in other studies varies significantly, from minimal values [31,32,33] to those exceeding 80% [43,44]. This variation is attributable not only to species diversity but also to the diversity of antioxidant types. High DPPH activity may be attributed to the substantial presence of organic molecules with good alcohol solubility, whereas low values suggest the absence of antioxidant molecules or the presence of alcohol-insoluble molecules. The proposed pretreatment methodology offers a partial solution to the issue of antioxidant type in the investigation of species [45].

In contrast, the values obtained from the ABTS analysis unequivocally demonstrated the substantial antioxidant activity of the extracts, which can be further optimized using the RSM methodology. The significantly elevated values observed here suggest the presence of antioxidant-like compounds within the extract. These compounds are likely molecules that are either water-soluble or bound to other macromolecules, such as polysaccharides, that are soluble in the extract [46]. While the DPPH analysis reflects the quantity of low-molecular-weight antioxidants in the extract, the most bioactive compounds in solution are likely polysaccharide-bound antioxidant molecules. Among the wild mushrooms examined, *Tricholoma equestre* exhibited the highest ABTS scavenging activity, with a maximal value of 86%, whereas *Suillus luteus* displayed the lowest activity, with a maximal value of 48%, as presented in Table 10.

The optimization of antioxidant activity, as assessed by ABTS cation radical scavenging, demonstrated the potential to correlate the polysaccharide content in the extract with its antioxidant efficacy. The maximum ABTS scavenging activity was achieved under the conditions of approximately 85 °C, 5 h, and a 120 LS ratio for *S. luteus*; 75 °C, 4 h, and a 95 LS ratio for *Tricholoma equestre*; and 90 °C, 5 h, and a 50 LS ratio for *Hydnum repandum* (Table 7). Although the optimal parameters for both optimizations were not identical, they revealed a shared parameter space conducive to extracting both high polysaccharide content and significant antioxidant activity (Table 2 and Table 7). This results in a synergistic bioactive effect of the extract, characterized by a high concentration of dietary fiber (polysaccharide) and enhanced radical scavenging capability.

The third method employed for antioxidant analysis, hydroxyl radical scavenging, did not succeed in optimizing scavenging activity in *Suillus luteus* and *Tricholoma equestre*, resulting in a suboptimal model (R^2^ < 0.55). Conversely, the model for *Hydnum repandum* was successfully developed, achieving an R^2^ of 0.85. This phenomenon can be attributed to the high reactivity of hydroxyl radicals, which may be quenched by any macromolecules present in the solution (extract). The assay’s lower stability and reproducibility can be ascribed to the rapid decomposition and high sensitivity to experimental conditions, such as temperature and sample composition [47,48]. This hypothesis is supported by comparing the hydroxyl radical scavenging with the TCC content in the samples. For both *Suillus luteus* and *Tricholoma equestre*, the TCC was high, indicating a substantial concentration of carbohydrate macromolecules in solution, which may facilitate the quenching of hydroxyl radicals, leading to instability during analysis. In contrast, the TCC obtained from *Hydnum repandum* is lower, resulting in less frequent quenching phenomena and greater stability during analysis, thereby yielding an accurate model. The optimal parameters for *Hydnum repandum* are presented in Table 8 and are similar to those observed for TCC, which may confirm the explanation of OH radical quenching on polysaccharide chains in the extract. A detailed interpretation of the results from the antioxidant activity analyses and optimization indicates that temperature plays a significant role in the solubility of antioxidant compounds (or antioxidants bound to macromolecules) in the mushroom extracts; however, it must be controlled to prevent the decomposition of antioxidants. Additionally, the water ratio regulates mass transfer during extraction, facilitating the diffusion of antioxidant compounds. Extraction time may not be the dominant factor under optimal temperature and water ratio conditions, as prolonged time may lead to the degradation of sensitive antioxidants. Although all wild mushrooms exhibit a similar trend, their diverse compositions result in varying observations under different conditions.

In all cases of optimization processes, Statistica 13.1 (StatSoft, Kraków, Poland) software was employed in order to find the ideal conditions. They were determined by optimizing the desirability of the responses. The desirability was maximized with the highest TCC concentration and the highest DPPH and ABTS values, while minimizing the RS parameter. These optimal conditions were then applied to the extraction process, and the responses were assessed and confirmed through a predefined procedure. Under these conditions, the experimental results matched the predicted values, with a coefficient of variation (CV) as indicated in Table 9.

## 3. Materials and Methods

### 3.1. Materials

*Suillus luteus* (L.) Roussel, as well as *Tricholoma equestre* (L.) P. Kumm., was acquired from a local market in Krakow, Poland, as fresh fruiting bodies. *Hydnum repandum* L. was collected in a mixed forest stand (*Fagus sylvatica*—50%, *Abies alba* Mill—40% and *Picea abies* (L.) H.Karst—10%. The altitude range of the forest is 400 to 680 m ASL) around Małastów village in Poland (Malopolskie Voivodeship, in the Administrative District of Gorlice, in the Municipality of Sekowa). Each mushroom species was bought/collected in autumn 2025 (September) in two batches (5 kg each) that were mixed together to standardize the experimental sample. The chemical reagents, such as DPPH (2,2-Diphenyl-1-picrylhydrazyl), ABTS (2,2′-azino-bis(3-ethylbenzothiazoline-6-sulfonic acid)), methanol, persulfate, K_2_HPO_4_, KH_2_PO_4_, aluminum chloride, potassium acetate, phenol, sulphuric acid, DNS (3,5-dinitrosalicylic acid) reagent, D-glucose, NaOH, and potassium sodium tartrate, were sourced from Sigma-Aldrich (Poznan, Poland) and utilized as pure for analysis.

### 3.2. Method

#### 3.2.1. Experimental Design

The experiment was optimized utilizing Response Surface Methodology (RSM) to enhance the extraction of polysaccharides (TCC), reducing sugar content, and antioxidant activity (DPPH, ABTS, and H_2_O_2_) for each mushroom species individually. A Box–Behnken experimental design was implemented to develop a second-order polynomial model for the extraction process. All responses were assessed based on the combined influence of three parameters: extraction time (X3), temperature (X2), and solvent-to-solid ratio (X1), each examined at three levels corresponding to low, medium, and high. The Box–Behnken matrix comprised 15 experiments for each mushroom species, including triplicates at the central points. The variables and experimental design levels employed in the Box–Behnken design are detailed in Table 10. The proposed methodology facilitates the evaluation of factor effects and the optimization of the influence of independent variables on the extraction process. Through the application of multiple regression analysis to the experimental data, the optimal values for extraction time, process temperature, and solvent-to-solid ratio were determined [14,49]. A second-order polynomial model equation was utilized to depict the generalized mathematical quadratic response surface. The equation for this model is as follows [50]:(1)$$y=\;\beta_{0}+\sum_{i=1}^{k}\beta_{i}x_{i}+\left(\sum_{i=1}^{k-1}\sum_{i=j=1}^{k}\beta_{ji}x_{j}x_{i}\right)+\sum_{i=1}^{k}\beta_{ii}x_{i}^{2}+\varepsilon$$
where y is the response value; k is the number of variables (for applied experimental design the number of independent variables, k = 3); β_0_ is defined as the constant effect; β_i_ is the linear regression coefficient; β_ii_ is the quadratic regression coefficient; β_ji_ is the interaction regression between the parameters; x_i_ and x_j_ are represented by the levels of the independent coded variables; ε is the error.

In the presented study, the Statistica 13.1 software (Statsoft, Poland) was employed to make all calculations and analyze results.

**Table 10 molecules-30-04647-t010:** Experiment conditions and observed results from carbohydrates extracted from wild mushrooms.

Run		Independent Variables			TCC, mg/g	RS, µmol/mL	DPPH	ABTS	H_2_O_2_
		Time (X3), h	Temperature (X2), °C	Liquid/Solid Ratio (X1) *v*/*w*			%		
1	*Suillus luteus* (L.) Roussel	3	70	100	6.246 ± 0.036	1.967 ± 0.019	4.20 ± 0.25	36.52 ± 3.41	33.33 ± 1.48
2		3	80	50	5.826 ± 0.033	2.985 ± 0.043	7.36 ± 0.21	48.36 ± 0.76	26.80 ± 1.62
3		3	80	150	6.777 ± 0.231	1.733 ± 0.033	5.67 ± 1.04	32.49 ± 2.39	22.98 ± 0.56
4		3	90	100	7.310 ± 0.037	2.010 ± 0.020	8.81 ± 0.21	44.21 ± 4.38	46.93 ± 0.56
5		4	70	150	9.233 ± 0.334	2.017 ± 0.073	7.24 ± 0.36	33.00 ± 2.65	8.74 ± 0.97
6		4	90	50	6.214 ± 0.034	2.917 ± 0.081	9.41 ± 0.42	41.56 ± 1.51	35.28 ± 3.68
7		4	90	150	11.837 ± 0.380	2.128 ± 0.056	4.23 ± 0.22	45.14 ± 4.05	8.51 ± 1.06
8		4	80	100	5.499 ± 0.782	2.744 ± 0.121	7.11 ± 0.21	44.02 ± 2.08	12.01 ± 1.19
9		4	70	50	6.942 ± 0.749	5.446 ± 0.621	5.55 ± 0.42	41.41 ± 2.10	27.83 ± 2.24
10		5	70	100	8.450 ± 0.074	2.393 ± 0.086	11.34 ± 0.75	46.28 ± 3.23	4.13 ± 0.21
11		5	80	50	3.851 ± 0.061	4.447 ± 0.106	12.12 ± 0.26	46.25 ± 3.60	70.21 ± 8.71
12		5	80	150	28.242 ± 0.379	2.245 ± 0.106	0.36 ± 0.02	43.83 ± 4.21	19.39 ± 1.63
13		5	90	100	12.798 ± 0.589	2.794 ± 0.475	10.25 ± 0.55	47.04 ± 2.34	7.37 ± 1.39
14		80	100	100	5.771 ± 0.432	2.744 ± 0.631	7.10 ± 0.20	43.32 ± 1.90	11.70 ± 1.06
15		80	100	100	5.228 ± 0.816	2.794 ± 0.085	7.11 ± 0.21	43.58 ± 2.31	11.65 ± 0.99
1	*Tricholoma equestre* (L.) P. Kumm.	3	70	100	7.778 ± 0.064	6.994 ± 0.089	3.56 ± 0.38	86.57 ± 0.46	32.38 ± 0.92
2		3	80	50	3.921 ± 0.051	11.692 ± 0.035	5.13 ± 0.48	49.70 ± 3.94	54.29 ± 0.95
3		3	80	150	11.766 ± 0.255	6.559 ± 0.020	8.94 ± 0.89	65.56 ± 0.46	16.19 ± 0.94
4		3	90	100	7.867 ± 0.115	13.489 ± 0.035	3.28 ± 0.34	53.54 ± 1.85	7.62 ± 0.98
5		4	70	150	11.705 ± 0.191	6.935 ± 0.247	4.29 ± 0.31	64.24 ± 0.61	55.87 ± 2.91
6		4	90	50	3.928 ± 0.051	11.105 ± 0.054	3.44 ± 0.13	61.43 ± 1.17	39.05 ± 1.90
7		4	90	150	7.905 ± 0.004	11.093 ± 0.176	11.31 ± 0.63	60.20 ± 0.17	64.44 ± 1.45
8		4	80	100	7.854 ± 0.009	7.640 ± 0.282	2.38 ± 0.21	83.33 ± 0.30	42.54 ± 4.89
9		4	70	50	7.922 ± 0.01	7.886 ± 0.282	6.28 ± 0.58	80.81 ± 0.35	27.30 ± 1.98
10		5	70	100	11,620 ± 0.006	7.546 ± 0.300	5.55 ± 0.28	79.70 ± 1.21	51.43 ± 2.86
11		5	80	50	11.780 ± 0.191	11.844 ± 0.405	5.13 ± 0.46	51.82 ± 3.15	70.16 ± 3.34
12		5	80	150	7.870 ± 0.163	7.182 ± 0.176	4.61 ± 0.48	65.15 ± 0.61	26.03 ± 2.40
13		5	90	100	3.947 ± 0.081	11.046 ± 0.370	6.63 ± 0.58	72.42 ± 0.00	94.29 ± 1.90
14		4	80	100	11.833 ± 0.287	7.640 ± 0.282	2.42 ± 0.20	83.38 ± 0.23	40.48 ± 3.33
15		4	80	100	7.915 ± 0.1711	7.640 ± 0.282	2.27 ± 0.04	83.18 ± 0.15	43.65 ± 3.10
1	*Hydnum repandum* L.	3	70	100	16.571 ± 0.247	7.872 ± 0.343	6.46 ± 0.23	6.81 ± 0.67	45.87 ± 2.90
2		3	80	50	8.405 ± 0.413	9.778 ± 0.367	6.46 ± 0.64	19.40 ± 1.29	48.54 ± 3.36
3		3	80	150	23.543 ± 0.428	7.575 ± 0.272	7.28 ± 0.35	1.73 ± 0.19	43.45 ± 1.46
4		3	90	100	15.814 ± 0.258	8.547 ± 0.296	4.64 ± 0.41	12.80 ± 1.29	31.72 ± 5.69
5		4	70	150	23.270 ± 0.300	8.111 ± 0.382	4.72 ± 0.51	7.74 ± 0.62	45.63 ± 4.12
6		4	90	50	7.879 ± 0.0415	12.538 ± 2.615	8.78 ± 0.87	65.43 ± 2.19	57.52 ± 4.78
7		4	90	150	23.358 ± 0.053	8.289 ± 0.213	3.99 ± 0.12	25.75 ± 1.04	25.24 ± 0.97
8		4	80	100	15.784 ± 0.194	9.183 ± 1.109	8.14 ± 1.31	17.34 ± 2.17	69.42 ± 2.80
9		4	70	50	7.937 ± 0.101	9.838 ± 0.661	7.51 ± 0.23	26.01 ± 1.38	27.51 ± 7.16
10		5	70	100	15.768 ± 0.233	7.892 ± 0.342	3.79 ± 0.30	30.03 ± 2.75	13.59 ± 1.68
11		5	80	50	8.114 ± 0.4087	11.565 ± 1.159	4.38 ± 0.55	60.84 ± 2.18	11.65 ± 0.97
12		5	80	150	23.407 ± 0.271	7.098 ± 0.000	1.79 ± 0.11	28.64 ± 2.39	65.29 ± 0.93
13		5	90	100	15.414 ± 0.041	6.860 ± 0.000	0.94 ± 0.81	16.87 ± 4.11	20.71 ± 3.12
14		4	80	100	15.836 ± 0.241	8.170 ± 0.000	8.63 ± 1.00	16.61 ± 1.66	69.26 ± 3.41
15		4	80	100	15.731 ± 0.166	10.195 ± 0.000	8.47 ± 1.24	18.42 ± 1.08	70.55 ± 2.02

To evaluate the influence of each factor (temperature, time, and solid to liquid ratio) on the response variable, an analysis of variance (ANOVA) was performed at a 95% confidence level. The regression coefficient (R^2^), *p*-value of the regression model, and *p*-value of the lack of fit (LOF) were examined to assess the accuracy of the regression model (see Appendix A). Optimal conditions were determined and validated based on the maximum total carbohydrate content (TCC), the quantity of reducing aldehyde groups in the extract (reducing sugars—RSs), and antioxidant activities (DPPH, ABTS, and H_2_O_2_) separately, as determined using RSM. The experimental values were compared with the predicted values using the coefficient of variation (CV%) to validate the model. The validity and adequacy of the predictive extraction (*n* = 3) were verified using a two-sided *t*-test (*p* = 0.05) to compare predictions with observed values.

#### 3.2.2. Extraction Process

The study examined the hot water extraction process by optimization using three parameters: a liquid-to-solid ratio from 50:1 to 150:1 *v*/*w*, mL/g (X1), temperatures ranging from 70 to 90 °C (X2), and extraction time between 3 and 5 h (X3) according to the preliminary study as well as procedure described in our previous study on extraction of carbohydrates from *Agaricus bisporus* [14]. These parameters were chosen based on initial research as well as already published data for the extraction of polysaccharides from *Agaricus bisporus*. The dependent variables measured included total carbohydrate content (TCC, mg/g), the concentration of free aldehyde groups (reducing sugars—RSs, µmol/mL) determined using DNS, and the percentage of radical scavenging ability evaluated through DPPH, H_2_O_2_ (hydroxyl radical scavenging), and ABTS.

The extraction procedure comprised several methodical steps as described in Khalil et al.’s work [14]. Initially, fresh mushrooms were thoroughly cleansed with cold tap water and subsequently dried at ambient temperature (20 °C) for a duration of 5 h. To remove low-molecular-weight compounds, including simple carbohydrates, the samples were agitated at 100 rpm with pure methanol (liquid–solid ratio 1:100 *v*/*w*) at room temperature (20 °C) for 2 h. Post-pretreatment, the samples were filtered through a paper filter, and the solid pretreated samples were allowed to dry for 24 h at room temperature [51]. Following the drying as well as pretreating process, the mushrooms were immediately frozen at −18 °C and then subjected to freeze-drying using a benchtop freeze-drying system (FreeZone 6 Liter Benchtop Freeze-Dry System; Labconco, Kansas City, MO, USA). The sample was then ground for 5 min utilizing a laboratory grinder.

Following, 1.00 g (based on dry matter) of freeze-dried sample of each mushroom species was mixed with different quantities of deionized water (X1) under various temperatures (X2) and extraction time (X3), with constant stirring, as outlined in Table 4. After the extraction process was completed, the sample was subjected to centrifugation at 9000 rpm for 10 min using a high-speed brushless centrifuge (MPW-350R, MPW, Warszawa, Poland). The supernatant was then collected and stored at −18 °C for later analysis. Each extraction run was performed six times for each sample.

#### 3.2.3. Total Carbohydrate Content Analysis

The Dubois method was employed for the analysis of TCC content [52,53]. In this method, 1.0 mL of a 5% phenol solution was combined with 1.0 mL of the sample extract in a test tube, followed by the addition of 5.0 mL of concentrated sulfuric acid (96%). The mixture was then allowed to react for 10 min, cooled to room temperature (20 °C) for 20 min, and thoroughly mixed. For the colorimetric analysis, a UV-Vis spectrophotometer (UV-VIS Dual Beam UVS-2800, Labomed Inc., Los Angeles, CA, USA) was used at a wavelength of 490 nm, with water and the reagents (phenol solution and sulfuric acid) serving as the blank. A standard curve was established using a glucose solution, and the results were expressed in milligrams of glucose. Each sample was analyzed in triplicate to ensure precision.

#### 3.2.4. Analysis of Reducing Sugar Content

The quantification of reducing sugar content was conducted utilizing the 3,5-dinitrosalicylic acid (DNS) method, which is predicated on the presence of free aldehyde groups in carbohydrate molecules within the extracts [54]. The optimization of the reducing sugars (designated as -CHO group concentration in the extracts) was aimed at minimizing their value, thereby facilitating the formation of longer polysaccharide chains in the solution. The analytical procedure was executed as follows: 1.0 mL of the extract was mixed with 0.75 mL of DNS solution and 3.25 mL of distilled water. This mixture was subjected to heating at 80 °C in a water bath for 10 min, followed by cooling in water for 20 min to reach room temperature (20 °C). The reaction was subsequently analyzed using UV-Vis spectroscopy (Labomed Inc., Los Angeles, CA, USA) at a wavelength of 540 nm. A blank sample was prepared in an identical manner, substituting water for the sample. A glucose solution was employed to construct a standard curve, and the results were expressed as µmol/mL of glucose. All samples were analyzed in triplicate.

#### 3.2.5. DPPH Radical Scavenging Activity

To assess the extract’s potential for scavenging free radicals, a method utilizing stable synthetic radicals of DPPH was employed [55]. A 0.1 mL aliquot of the extract was introduced into a 3.9 mL DPPH solution (prepared from a 1.12 mM solution in MeOH and diluted with methanol to achieve an absorbance of 0.8 ± 0.02 at 515 nm). The mixture was vigorously vortexed for 30 s and subsequently left in the dark at room temperature for 30 min. The decolorization of DPPH was quantified using a UV-Vis spectrophotometer (Labomed Inc., USA) at 515 nm, and the DPPH free radical scavenging activity (%) was determined as the percentage reduction in absorbance. The blank sample was prepared by using a methanolic dilution of DPPH (0.1 mL of pure methanol was added to a 3.9 mL solution of DPPH).

#### 3.2.6. ABTS Radical Scavenging Activity

The ABTS+ solution was prepared by combining equal volumes of ABTS solution (7 mM) and potassium persulfate solution (2.4 mM), followed by a 12–16 h incubation at room temperature, protected from light exposure. Subsequently, the solution was diluted with distilled water to achieve an initial absorbance of 0.70 ± 0.02 at 734 nm. The reaction was initiated by introducing 50 µL of the sample into 950 µL of water, followed by the addition of 1 mL of the ABTS+ solution. This mixture was then incubated for 7 min at room temperature. The reduction in absorbance was measured at 734 nm using a UV-Vis spectrophotometer (Labomed Inc., USA), and the ABTS+ scavenging activity was quantified by the percentage decrease in absorbance. Equal volumes of ABTS and water were used as controls [56].

#### 3.2.7. Hydroxyl Radicals Scavenging Activity Assay

The hydroxyl radical scavenging activity was assessed by employing hydroxyl radicals generated from hydrogen peroxide solution. The procedure was performed as follows. 0.6 mL of extract was mixed together with 1.8 mL of 50 mM phosphate buffer (pH = 7.4) and 3.6 mL of a 2 mM hydrogen peroxide solution. The resulting mixture was vortexed and allowed to stand for 10 min at room temperature. The reduction in absorbance was measured at 230 nm using a UV-Vis spectrophotometer (Labomed Inc., USA), and the scavenging activity (%) was calculated as the percentage decrease in absorbance. A blank sample was prepared by mixing 0.6 mL of distilled water with a 3.6 mL working solution, specifically H_2_O_2_ in buffer [55].

#### 3.2.8. Validation of the Model

The optimal conditions for polysaccharide extraction, encompassing all variables (EH, ET, and LS), were evaluated to achieve the highest carbohydrate content while minimizing the content of free aldehyde groups and maximizing antioxidant activity, as assessed using ABTS and hydroxyl radical scavenging assays. These conditions were determined through Response Surface Methodology (RSM), and the experimental values obtained were compared with those predicted by the model to verify their accuracy. All responses were re-evaluated under optimized extraction conditions to confirm the results.

## 4. Conclusions

Research has unequivocally demonstrated that edible wild mushrooms, often perceived as having lower culinary quality, can serve as a significant source of bioactive polysaccharides with antioxidant properties. The extraction of these fractions using hot water has been shown to have positive outcomes by means of polysaccharide yield and is considered one of the most environmentally sustainable methods, as it does not involve the use of harmful chemicals. Although the yield of polysaccharide extraction from wild mushrooms is lower compared to cultivated varieties, their overall availability renders them a promising source of bioactive compounds. Optimization of water extraction parameters has revealed the impact of the high liquid-to-solid (LS) ratio on extraction efficiency, particularly in maximizing total carbohydrate content (150:1 for *Suillus luteus*; 149.76 for *Tricoloma equestre* and 149.89 for *Hydnum repandum*), while temperature and time parameters varied among the species studied (5 h, 89.92 °C for *Suillus luteus*; 3 h, 70.01 °C for *Tricholoma equestre* and 3.98 h, 70.07 °C for *Hydnum repandum*). It was also noted that optimization can be achieved by maximizing TCC and minimizing reducing sugars (RSs), facilitating the extraction of polysaccharides with longer macromolecular chains. The observed bioactivity of the extracts is primarily attributed to their antioxidant activity, which is probably based on water-soluble antioxidants either bound to or hydrolyzed from macromolecular compounds (polysaccharides). The optimal condition for maximizing the antioxidant activity of aqueous extracts (ABTS method) was found to be as follows: (123.68 LS ratio, 5 h, 86.63 °C for *Suillus luteus*; 95.27 LS ratio, 4.05 h, 73.87 °C for *Tricholoma equestre* and 50.01 LS ratio, 5.00 h, 89.98 °C for *Hydnum repandum*). Based on these conclusions, subsequent research will focus on a detailed structural investigation and the potential application of crude extracts in novel food product formulations.

## Figures and Tables

**Table 1 molecules-30-04647-t001:** Regression coefficient (β) and coefficient of determination (R^2^) values of the predicted models for carbohydrate extraction from wild mushrooms.

	*Suilus luteus*	*Tricholoma equestre*	*Hydnum repandum*
Intercept			
β_0_	5.499	7.905	15.784
Linear			
β_1_	8.314	5.917	15.418
β_2_	1.822	1.975	−0.378
β_3_	6.766	*p* > 0.01	−0.407
Quadratic			
β_11_	5.530	1.862	*p* > 0.01
β_22_	*p* > 0.01	1.845	*p* > 0.01
β_33_	5.819	−1.939	*p* > 0.01
Mixed			
β_12_	*p* > 0.01	−3.809	*p* > 0.01
β_13_	11.719	*p* > 0.01	*p* > 0.01
β_23_	*p* > 0.01	*p* > 0.01	*p* > 0.01
R^2^	0.878	0.872	0.997

**Table 2 molecules-30-04647-t002:** Optimal and predicted value for carbohydrate content from wild mushrooms (maximal values in the investigated range).

		Maximal, mg/g	±95% CI for Mean	±95% CI for 99% Population
*Suillus luteus*				
Time, h	5.00	27.446	25.311–29.581	19.131–35.759
Temperature, °C	89.92			
Ratio, *v*/*w*	150,00			
*Tricholoma equestre*				
Time, h	3.98	12.634	12.480–14.788	9.018–18.250
Temperature, °C	70.07			
Ratio, *v*/*w*	149.89			
*Hydnum repandum*				
Time, h	3.00	23.921	23.761–24.081	22.913–24.929
Temperature, °C	70,01			
Ratio, *v*/*w*	149.76			

**Table 3 molecules-30-04647-t003:** Regression coefficient (β) and coefficient of determination (R^2^) values of the predicted models for reducing sugars extracted from wild mushrooms.

	*Suillus luteus*	*Tricholoma equestre*	*Hydnum repandum*
Intercept			
β_0_	2.744	7.640	8.695
Linear			
β_1_	−1.903	−2.690	−3.161
β_2_	−0.478	4.343	*p* > 0.01
β_3_	0.796	*p* > 0.01	*p* > 0.01
Quadratic			
β_11_	0.959	*p* > 0.01	2.210
β_22_	*p* > 0.01	2.064	−1.592
β_33_	−0.743	2.193	*p* > 0.01
Mixed			
β_12_	1.350	*p* > 0.01	−1.261
β_13_	−0.475	*p* > 0.01	−1.132
β_23_	*p* > 0.01	−1.498	*p* > 0.01
R^2^	0.876	0.840	0.763

**Table 4 molecules-30-04647-t004:** Optimal value for reducing sugar content extracted from wild mushrooms (minimal values in the investigated range).

		Predicted Response		
	Reducing Sugars	Minimal, µmol/mL	±95% CI for Mean	±95% CI for 99% Population
		*Suilus luteus*		
Time, h	3.05	1.383	1.046–1.721	0.219–2.786
Temperature, °C	70.11			
Ratio, *v*/*w*	147.15			
		*Tricholoma equestre*		
Time, h	4.98	6.207	5.446–6.967	2.591–9.822
Temperature, °C	88.32			
Ratio, *v*/*w*	141.10			
		*Hydnum repandum*		
Time, h	3.76	6.037	5.389–6.686	2.229–9.846
Temperature, °C	74.72			
Ratio, *v*/*w*	133.80			

**Table 5 molecules-30-04647-t005:** Regression coefficient (β) and coefficient of determination (R^2^) values of the predicted models for antioxidant activities of carbohydrates extracted from wild mushrooms.

	*Suilus luteus*			*Tricholoma equestre*			*Hydnum repandum*		
	DPPH	ABTS	H_2_O_2_	DPPH	ABTS	H_2_O_2_	DPPH	ABTS	H_2_O_2_
Intercept									
β_0_	7.117	44.018	11.964	2.358	83.314	42.245	8.208	17.337	69.417
Linear									
β_1_	−4.299	−5.776	−23.739	2.292	*p* > 0.01	*p* > 0.01	−2522	−25.220	8.213
β_2_	*p* > 0.01	5.296	*p* > 0.01	1.245	−15.930	*p* > 0.01	−0.854	13.218	*p* > 0.01
β_3_	1.854	5.821	*p* > 0.01	*p* > 0.01	*p* > 0.01	32.857	−3.365	22.827	−14.525
Quadratic									
β_11_	−3.052	−4.702	18.643	5.167	−31.643	*p* > 0.01	*p* > 0.01	26.410	−15.476
β_22_	1.805	*p* > 0.01	*p* > 0.01	2.776	*p* > 0.01	*p* > 0.01	−3.016	*p* > 0.01	−45.406
β_33_	*p* > 0.01	*p* > 0.01	24.344	2.019	−18.873	*p* > 0.01	−5699	*p* > 0.01	−36.599
Mixed									
β_12_	−3.216	5.471	*p* > 0.01	4.926	7.669	*p* > 0.01	*p* > 0.01	−11.550	−25.202
β_13_	−2.835	6.606	−26.274	−2.164	*p* > 0.01	*p* > 0.01	−1.707	*p* > 0.01	30.369
β_23_	−5.383	−3.506	*p* > 0.01	*p* > 0.01	12.879	33.810	*p* > 0.01	−10.116	11.523
R^2^	0.821	0.719	0.540	0.824	0.845	0.536	0.854	0.821	0.854

**Table 6 molecules-30-04647-t006:** Optimal value for DPPH scavenging activity of carbohydrates extracted from wild mushrooms (maximal values in the investigated range).

		Maximal, %	±95% CI for Mean	±95% CI for 99% Population
*Suilus luteus*				
Time, h	4.90	12.606	11.115–14.097	6.640–18.572
Temperature, °C	88.17			
Ratio, *v*/*w*	53.29			
*Tricholoma equestre*				
Time, h	3.02	12.048	10.465–13.631	6.964–17.132
Temperature, °C	89.89			
Ratio, *v*/*w*	149.42			
*Hydnum repandum*				
Time, h	3.85	9.152	8.408–9.896	5.154–13.150
Temperature, °C	80.58			
Ratio, *v*/*w*	50.00			

**Table 7 molecules-30-04647-t007:** Optimal value for ABTS scavenging activity of carbohydrates extracted from wild mushrooms (maximal values in the investigated range).

		Predicted Response		
		Maximal, %	±95% CI for Mean	±95% CI for 99% Population
*Suilus luteus*				
Time, h	5.00	48.450	45.932–50.964	36.414–60.485
Temperature, °C	86.63			
Ratio, *v*/*w*	123.68			
*Tricholoma equestre*				
Time, h	4.05	87.701	84.033–91.370	66.973–108.609
Temperature, °C	73.87			
Ratio, *v*/*w*	95.27			
*Hydnum repandum*				
Time, h	5.00	65.064	55.057–75.071	30.945–99.184
Temperature, °C	89.98			
Ratio, *v*/*w*	50.01			

**Table 8 molecules-30-04647-t008:** Optimal value for H_2_O_2_ scavenging activity of carbohydrates extracted from wild mushrooms (maximal values in the investigated range).

		Predicted Response		
		Maximal, %	±95% CI for Mean	±95% CI for 99% Population
*Hydnum Repandum*				
Time, h	3.86	70.237	64.456–76.017	39.322–101.151
Temperature, °C	79.56			
Ratio, *v*/*w*	108.42			

**Table 9 molecules-30-04647-t009:** The validation of predicted values using experimental data at optimal extraction conditions of carbohydrates from wild mushrooms.

Dependent Variables	Predicted Value	Experimental Value	%CV
*Suillus luteus*			
TCC ^1^, mg/g	27.422	28.536	14.56
RS ^2^, µmol/mL	1.383	1.736	9.78
DPPH ^3^, %	12.606	12.309	10.05
ABTS ^4^, %	48.450	49.871	6.85
*Tricholoma equestre*			
TCC, mg/g	12.634	12.116	11.63
RS, µmol/mL	6.207	6.859	11.25
DPPH, %	12.048	11.938	13.39
ABTS, %	87.701	86.972	7.98
*Hydnum repandum*			
TCC, mg/g	23.921	23.890	1.84
RS, µmol/mL	6.037	6.199	10.84
DPPH, %	9.152	9.620	12.00
ABTS, %	65.064	63.92	13.14
H_2_O_2_ ^5^, %	70.237	72.82	9.98

^1^ TCC—Total Carbohydrates Content. ^2^ RS—Reducing Sugars. ^3^ DPPH—DPPH Scavenging activity. ^4^ ABTS—ABTS Scavenging activity. ^5^ H_2_O_2_—H_2_O_2_ Scavenging activity.

## Data Availability

The data are included in this article.

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
