# Peer review of "Optimized Extraction of Bioactive Polysaccharides from Wild Mushrooms: Toward Enhanced Yield and Antioxidant Activity"

_molecules, 2025, doi:10.3390/molecules30234647_

Round 1
Reviewer 1 Report
Comments and Suggestions for Authors
This manuscript investigates the optimization of hot water extraction of bioactive polysaccharides from three wild mushroom species (Suillus luteus, Tricholoma equestre, and Hydnum repandum), employing Response Surface Methodology to examine the effects of liquid-to-solid ratio, temperature, and time on total carbohydrate content and antioxidant activity. The research topic has certain scientific significance and application value. Minor revision is recommended before publication.
Specific Comments:
Line 2-3: The phrase "Enhanced Yield and Functionality" in the title is vague. It is recommended to specify what "functionality" refers to (e.g., antioxidant activity).
Line 12-13: Please check throughout the manuscript whether Trichloma equestre is misspelled.
Line 24-25: The conclusion "confirming the LS ratio as critical for extracting longer polysaccharide chains" is overly absolute. In fact, the reduction in RS values can only indirectly suggest that the degree of polymerization may be higher, but cannot directly prove chain length.
Line 35-36: The statement "Mushrooms, representing the third domain of life on Earth" is inaccurate. Please verify whether fungi constitute the third "domain" of life.
Line 42-44: Market data cites 2024 figures (Line 42) with projections to 2033 (Line 44), but the reliability of references 5 and 6 needs verification. Additionally, these data have weak relevance to this study.
Line 72-76: The rationale for selecting these three wild mushroom species in the introduction is insufficient. Why were these specific mushrooms chosen? More compelling scientific justification is needed beyond being "common in forests."
Line 96-112: The discussion of T. equestre toxicity is overly lengthy and has weak relevance to the research topic. If this mushroom has toxicity concerns, why was it selected as a research subject? This requires clearer explanation.
Line 111-112: The inference that "it may serve as a suitable raw material for the extraction of safe polysaccharide fractions" lacks safety verification data.
Line 130-139: The statement of research objectives is too general. Clear hypotheses and specific research questions are missing.
Line 371-376: Suillus luteus and Tricholoma equestre were purchased from local markets, while Hydnum repandum was collected from the wild. This inconsistency in sample sources may affect comparability of results and requires explanation.
Line 373-376: The collection site description is overly detailed (altitude, tree species composition, etc.), but lacks critical information such as collection time, sample quantity, and identification methods.
Line 384-427: The experimental design section has missing information: the rationale for selecting Box-Behnken design is not stated; the basis for parameter ranges (50-150 LS ratio, 70-90°C, 3-5 hours) is unclear; why wasn't a broader parameter range used?
Line 449-451: "1.00 g freeze-dried sample...mixed with different quantities of deionized water" but what state of sample (fresh weight or dry weight) is the LS ratio in Table 10 based on?
Line 468-481: Regarding the DNS method for reducing sugar determination: heating conditions are 80°C for 10 minutes, but the standard DNS method typically uses boiling water bath—please explain why conditions were modified; result units are inconsistent between "µmol/mL" (Line 213, etc.) and "ppm" (Table 10).
Line 150-160: "significantly lower than that in the previously studied Agaricus bisporus" but no specific numerical comparison or statistical testing is provided.
Line 152-154: The maximum value for Suillus luteus is approximately 28.5 mg/g, and the lowest for Hydnum repandum is approximately 12 mg/g, but do these values fully correspond to the data in Table 10? Please clarify under which conditions these values were obtained.
Line 175-196: The explanation for differences in optimal conditions among species is overly simplified and lacks in-depth mechanistic analysis.
Line 178-179: Does "the optimal LS ratio for Agaricus bisporus not exceeding 120:1" align with the data in reference 17? Verification is needed.
Line 180-185: The statement about "more complicated internal structure of polysaccharides" lacks structural analysis data support.
Line 186-196: The explanation that only linear coefficients are significant for Hydnum repandum is insufficient. This may indicate the model is not applicable, rather than "mass transfer is the simplest."
Line 226-233: The association between Suillus luteus "slippery jack" name and water-binding capacity is too tenuous and lacks scientific basis.
Line 237-244: The R² value for the RS model (Hydnum repandum is only 0.763) is relatively low, indicating poor model fit.
Line 253 (Table 3): The interpretation of positive/negative signs of β coefficients is not sufficiently clear.
Line 257-263: The explanation that Tricholoma equestre requires higher temperature and longer time (polysaccharides are strongly bound) is speculative and lacks experimental evidence.
Line 265 (Table 4): The table title states "Optimal value for RS content (maximal values...)" but actually shows minimum values—the title is incorrect.
Line 275-278: The hypothesis that antioxidants may be bound to macromolecules requires experimental validation (e.g., determination after molecular weight fractionation).
Line 303 (Table 6): Under optimal conditions, the LS ratio for Suillus luteus is 53.29, while for Tricholoma equestre it is 149.42—this substantial difference is insufficiently explained.
Line 522-539: The claim of "highly effective" (Line 525) lacks quantitative comparison support; "most environmentally sustainable methods" (Line 526) requires life cycle assessment data support; the conclusion about "polysaccharide-bound antioxidants" (Line 535-536) is speculative and not experimentally verified; "subsequent research will focus on...detailed structural investigation" (Line 537-538) should be work completed in this study.
Author Response
The answer for the reviewer is attached as a separate file.

Reviewer 2 Report
Comments and Suggestions for Authors
The manuscript titled “Optimized Extraction of Bioactive Polysaccharides from Wild Mushrooms: Toward Enhanced Yield and Functionality” presents a well-conducted study in which the authors optimized aqueous extraction of polysaccharides from Suillus luteus, Tricholoma equestre, and Hydnum repandum using Response Surface Methodology. They analyzed key parameters, such as polysaccharide content, reducing sugar content, and antioxidant activity (DPPH, ABTS, and H₂O₂). The results are relevant and potentially valuable. However, I recommend that the authors enhance the discussion section by incorporating additional, recent and relevant studies on polysaccharide extraction from wild mushrooms. By discussing their results in a broader context, they could better highlight the novelty and significance of their work. After minor revision, I suggest the publication or the manuscript.
Abstract
Line 1: Rewrite “Article”.
Line 11: The authors should insert the optimized parameters and the main responses obtained.
Line 29: Insert the following keywords: Suillus luteus; Trichloma equestre; Hydnum repandum; aqueous extraction; antioxidant activity;
Results and Discussion
Line 168: Rewrite “Table 1. Regression coefficient (β) and coefficient of determination (R2) values of the predicted models for carbohydrate extraction from wild mushrooms.”.
Line 177: Insert the highest value mentioned.
Line 197: Rewrite “Table 2. Optimal and predicted value for carbohydrate content from wild mushrooms (maximal values in investigated range).”
Line 255: Rewrite “Table 3. Regression coefficient (β) and coefficient of determination (R2) values of the predicted models for reducing sugars extracted from wild mushrooms.”
Line 265: Rewrite “Table 4. Optimal value for reducing sugars content extracted from wild mushrooms (maximal values in investigated range).”.
Line 265: Rewrite in table 4 content “Reducing sugars” instead of “RS”.
Line 280: Rewrite “Table 5. Regression coefficient (β) and coefficient of determination (R2) values of the predicted models for antioxidant activities of carbohydrates extracted from wild mushrooms”.
Line 300-301: The authors should provide a reference that supportting this statement.
Line 303: Rewrite “Table 6. Optimal value for DPPH scavenging activity of carbohydrates extracted from wild mushrooms (maximal values in investigated range).”
Line 317: Rewrite “Table 7. Optimal value for ABTS scavenging activity of carbohydrates extracted from wild mushrooms (maximal values in investigated range).”
Line 332 and 333: Rewrite “(R2<0.55)” and “R2”.
Line 357: Rewrite “Table 8. Optimal value for H2O2 scavenging activity of carbohydrates extracted from wild mushrooms (maximal values in investigated range).”
Line 368: Rewrite “Table 9. The validation of predicted values using experimental data at optimal extraction conditions of carbohydrates from wild mushrooms.”.
Line 368: Insert a caption under table 9 informing the meaning of TCC, RS, and the DPPH, ABTS and H2O2 expressed values.
Line 143-368: Can the authors find other relevant studies besides from 17 and 35 to improve the discussion of the obtained results?
- Khalil, A.S.E.; Lukasiewicz, M. The Optimization of the Hot Water Extraction of the Polysaccharide-Rich Fraction from Agari-592 cus Bisporus. Molecules 2024, 29, 4783, doi:10.3390/molecules29194783.
- Badoni, P.; Siddiqui, S.A. Metamorphosis of Mushroom Production from Tradition to Automation. Discov. Appl. Sci. 2025, 7, 634 974, doi:10.1007/s42452-025-07517-w.
Material and methods
Line 419: Rewrite “R2”.
Line 431: Please revise the liquid-to-solid ratio. In Tables the authors used w/v and in the Material and Methods section v/w.
Line 440 and 443: Rewrite “al” and “w/v”.
Line 443 and 445: Insert the room temperature value.
Line 457: Rewrite “Table 10. Experiment conditions and observed results from carbohydrates extracted from wild mushrooms.”. In addition, insert a caption with the meaning of the abbreviations used in the table. The authors must decide between Reducing sugars or -CHO group to be equally in the manuscript.
Line 458: Insert the time used for each step in Dubois methodology.
Line 476: Insert the room temperature value.
Conclusion
Line 528: Insert the “yield of polysaccharide”.
Line 531 and 532: Insert the values for liquid-to-solid, total carbohydrate, temperature and time mentioned.
Line 536: Insert the values obtained for antioxidant activity of the extracts.
Line 547: Rewrite “Education”.
Author Response

(The authors gave the same response as above.)

Reviewer 3 Report
Comments and Suggestions for Authors
This manuscript investigates the extraction of polysaccharide-rich fractions from three wild mushroom species (Suillus luteus, Tricholoma equestre, Hydnum repandum) using hot-water extraction optimized through Response Surface Methodology (RSM).
The work contributes to the growing interest in natural polysaccharide resources, especially from underutilized wild mushrooms and the experimental design is well-structured.
It can benefit from more clarification regarding why these three specific species were chosen beyond availability. Consider enhancing the final paragraph of the Introduction to clearly articulate the novelty and scientific necessity of this study.
Line 431: Ensure consistency for the units (units appear as w/v and sometimes v/w)
You mention six extraction runs per sample. Please clarify whether it is biological replicates, technical replicates, or total experiment repetitions.
Given that the central objective of this work is the optimization of extraction parameters, I strongly recommend including the specific optimal temperature, time, and LS ratio values in both the abstract and the conclusion. These parameters are critical outputs of the RSM analysis and would significantly enhance the clarity and practical applicability of the study.
Author Response

(The authors gave the same response as above.)

Reviewer 4 Report
Comments and Suggestions for Authors
The study presents the use of a water extracting system for bioactive polysaccharides from three wild mushroom species. For it a RSM process was developed in order to optimize and validate the proposed conditions of independent variables.
I think it is well presented, justified and discussed. It includes an eco-friendly procedure for extraction and focusses the preparation of bioactive compounds from natural resources.
However, before acceptation, some performances ought to be carried out.
I would mention the following:
Abstract
Lines 15-16: … liquid-to-solid (LS) ratio …
Line 16: Indicate the value ranges to be addressed for the three independent or processing variables.
Include the optimized values for the three processing variables.
Keywords
Replace “wild mushrooms” with “wild” followed by the names of the three species.
Replace extraction with water extraction.
Include: wild; yield; reducing sugars; antioxidant properties.
Introduction
This section is somewhat long. Paragraphs related to carbohydrates and each of the mushroom species could be reduced.
Results and discussion
This section is well presented and discussed. Notably, it includes a validation step, which shows the convenience of the procedure proposed.
Material and methods
Were there different batches of initial mushrooms taken into account ? Or was just a single one employed ?
Some justification for the range values chosen for each independent variable ought to be provided.
Author Response

(The authors gave the same response as above.)
